# Influence of DETA on Thermal and Corrosion Protection Properties of GPTMS-TEOS Hybrid Coatings on Q215 Steel

Shuanqiang Yang [1,†], Zhenzhen Jia [2,†], Jinjia Xu [3] and Ruoyu Hong [2,*]

1 Institute of Industrial Technology, Fujian Jiangxia University, Fuzhou 350108, China; yangshq@fjjxu.edu.cn
2 College of Chemical Engineering, Fuzhou University, Fuzhou 350108, China
3 Department of Chemistry and Biochemistry, University of Missouri-St. Louis, St. Louis, MO 63121, USA; jxu@umsl.edu
* Correspondence: rhong@fzu.edu.cn
† These authors contributed equally to this work.

**Abstract:** High-performance coating could be used to protect steels in engineering. The GPTMS-TEOS hybrid coatings were successfully prepared using (3-glycidoxypropyl) trimethoxysilane (GPTMS) and tetraethylorthosilicate (TEOS) as reaction raw materials and diethylenetriamine (DETA) as both a curing agent and catalyst at room temperature. The hybrid coating contained amorphous $SiO_2$ and was characterized by means of X-ray diffraction (XRD), Fourier transform infrared spectroscopy (FT-IR), scanning electron microscopy (SEM), and transmission electron microscopy (TEM). The DETA content of the hybrid coating has a significant impact on the performance of the coating. As the DETA content increases, the thermal stability of the hybrid coating increases at 400–600 °C due to the production of more $SiO_2$ in the amine-rich state. However, the gelation time decreases dramatically, preventing the hybrid coating from better infiltrating the surface of the steel substrate. In addition, there are not enough silicon hydroxyl groups to bond with the hydroxyl groups on the surface of carbon steel and adhesion is significantly reduced. Therefore, hybrid coatings with a moderate DETA content (NH:epoxy ratio equivalent to 1:1) show the best corrosion resistance, with a third-order magnitude increase in corrosion resistance compared to that of carbon steel.

**Keywords:** GPTMS-TEOS; hybrid coatings; amorphous $SiO_2$; anti-corrosion coatings





## 1. Introduction

Metals have been regarded as important engineering materials due to their high strength, toughness, good ductility and workability [1]. However, the corrosion of metals is a major problem that plagues people. The corrosion of metals is generally defined as damage caused by a chemical or electrochemical reaction when metals are in continuously close contact with surrounding media (atmosphere, soil, seawater, fresh water, etc.). Once corrosion occurs, the metal is transformed from a monomer into some other undesirable species, resulting in rusting, perforation, cracking and brittleness. According to relevant statistics, the annual loss due to metal corrosion worldwide accounts for about 3.4% of the GDP (gross domestic product) of the year, and the direct economic loss caused by corrosion is as high as one trillion USD per year [2]. The hazards of metal corrosion not only cause economic losses, but also damage to metal-related equipment and components; therefore, it can cause many accidents and a decline in product quality, leading to plant stoppages, production shutdowns and even explosions and safety accidents [3,4].

Sol–gel technology has been widely used in the preparation of anti-corrosion coatings for metals [5,6]. The essential chemical reaction of the sol–gel method is the hydrolysis and condensation of alcohol salts, leading to the formation of M-O-M at low temperatures. The disadvantage of oxide films based on simple alkoxysilanes such as tetraethylorthosilicate (TEOS) is that they tend to break down during heat treatment. Therefore, the synthesis of organic–inorganic hybrids using the sol–gel method is a viable method for preparing

dense, and crack-free coatings without the necessity of post-deposition annealing at high temperatures are highly desirable. In recent years, hybrid coatings based on the organic functional silane R'Si(OR)$_3$ have been widely used for the corrosion protection of metals. The incorporation of the organic component R' in the sol–gel matrix can control the density and flexibility of the sol–gel network [7], thereby increasing the maximum achievable film thickness without rupture. Various methods have been used to enhance the protective properties of silane coatings. These can basically be divided into active protection methods and other methods that improve barrier properties or the adhesion of the coating. Some studies have shown that the introduction of prefabricated nanoparticles (such as SiO$_2$, Al$_2$O$_3$ [8] and CeO$_2$ [9]) or in situ-generated nanoparticles (such as ZrO$_2$ [10]) can also improve the corrosion protection properties of sol–gel coatings by enhancing the hybrid matrix. In addition, the barrier properties of sol–gel coatings can be improved through the careful selection of suitable precursors. The use of fluorine replacement [11] or long-alkyl-chain [12] silanes impart hydrophobicity to sol–gel coatings. Alkoxysilanes with reactive organic functional groups present a new method for the preparation of coatings with excellent corrosion protection properties by changing the order of the siloxane network [13] or by promoting the silane network and polymer binding [14,15]. Among the functional alkoxysilanes, (3-glycidoxypropyl) trimethoxysilane (GPTMS) is one of the most commonly used silanes for protective coatings. Several studies have used GPTMS with amino silanes [16] or triethylenetetramines to prepare hybrid protective coatings while forming organic–inorganic networks [17–19]. The overall properties of these coatings were investigated and the importance of protection for pretreated AZ31 Mg alloys was demonstrated [20]. It has been reported that the high degree of cross-linking of this organic–inorganic hybrid coating provides effective protection to the metal, and the silicon hydroxyl groups in the hybrid coating increase adhesion by binding to the hydroxyl groups on the metal surface, further increasing the protective properties [21,22]. Therefore, the development of high-performance organic–inorganic hybrid coatings for the corrosion protection of metals is of great significance that is worthy of intensive research. In this paper, we reported a simple and straightforward method to prepare GPTMS-TEOS hybrid coatings using sol–gel technology, GPTMS and TEOS as raw materials and diethylenetriamine (DETA) as a cross-linker and curing agent. A series of hybrid coatings with a varying the amount of DETA have been prepared (see Table 1 for specific hybrid coatings) and optimized, allowing us to systematically investigate their microscopic properties through scanning electron microscopy (SEM), transmission electron microscopy (TEM), X-ray diffraction (XRD), Fourier transform infrared spectroscopy (FTIR) and thermogravimetric analysis (TGA). The corrosion behavior of the steel electrode and steel electrodes coated with hybrid coatings were investigated via electrochemical impedance spectroscopy (EIS) in a 3.5 wt.% NaCl solution at room temperature. The study provides an informative guideline for the preparation of high-performance organic–inorganic hybrid coatings and reveals the corrosion protection mechanism of the coatings.

**Table 1.** DETA dosage for different hybrid coatings.

| Samples (R$_x$) | R$_{0.2}$ | R$_{0.4}$ | R$_{0.5}$ | R$_1$ | R$_2$ | R$_3$ |
|---|---|---|---|---|---|---|
| X = NH/epoxy | 0.2 | 0.4 | 0.5 | 1 | 2 | 3 |

This paper analyzes the literature of anti-corrosion materials and proposes that GPTMS-TEOS hybrid coatings were successfully prepared at room temperature with (3-glycidyloxypropyl) trimethoxysilane (GPTMS) and ethylorthosilicate (TEOS) as heavy-acting raw materials, and diethylenetriamine (DETA) as a curing agent and catalyst. X-ray diffraction (XRD), Fourier transform infrared spectroscopy (FT-IR), scanning electron microscopy (SEM) and transmission electron microscopy (TEM) were used to characterize the hybrid coating, and the DETA content of the hybrid coating containing amorphous SiO$_2$.

The thermal stability of hybrid coatings was studied and the best dosage of the curing agent (DETA) was obtained.

## 2. Preparation of Hybrid Coatings

### 2.1. Materials

Tetraethylorthosilicate (TEOS) and ethanol were purchased from Sinopharm Chemical Reagent Co., Ltd. (Shanghai, China) Diethylenetriamine (DETA) and (3-glycidoxypropyl) trimethoxysilane (GPTMS) were obtained from Aladdin Co., Ltd. (Shanghai, China). All the reagents were of an analytical grade and had not been further purified. Q215 steel was obtained from Dongguan Chuicheng Metal Material Co., Ltd. (Dongguan, China). Deionized water was prepared by the laboratory.

### 2.2. Preparation of GPTMS-TEOS Hybrid Coatings

A mixture of 0.03 mol GPTMS and 0.01 mol TEOS was added to 10 mL of ethanol and a quantity of water and stirred magnetically for 4 h at room temperature to prepare the silanol. DETA was added to the above hydrolyzed silane–sol in the amount listed in Table 1 and the reaction was carried out at room temperature for 30 min to produce the hybrid coatings. A Q215 steel electrode with an exposed area of 1 cm$^2$ was used as the test electrode. The steel electrodes were sandpapered, ultrasonically degreased with isopropyl alcohol and dried in a nitrogen atmosphere prior to the tests. Then, the prepared hybrid coating was applied to the surface of a Q215 steel electrode by brushing. After painting, the samples were cured at room temperature for 1 week and then dried in an oven at 50 °C for 1 week. The thickness of the resulting coating measured using an ultrasonic thickness gauge was approximately 80 nm.

### 2.3. Characterization

A field emission scanning electron microscope (FESEM, JSM-6700F, JEOL, Tokyo, Japan) and transmission electron microscope (TEM, TecnaiG220, FEI Company, Hillsborough, OR, USA) were used for the morphological analysis of the hybrid coatings. The functional groups and phase structures of the hybrid coatings were characterized using a Fourier transform infrared spectrometer (FT-IR, Nicolet Avatar 360, Nicolet Company, Hillsboro, OR, USA) and X-ray powder diffractometer (XRD, X′ PertPro, PANalytical Company, Almelo, The Netherlands), respectively. The effect of DETA content on the thermal stability of the hybrid coating was investigated via thermogravimetric analysis (TGA, */STA449C/6/G, NETZSCH Corporation, Selb, Germany) in a temperature range of room temperature to 700 °C in an N$_2$ atmosphere. The corrosion behavior of the steel electrode and steel electrodes coated with hybrid coatings were investigated via electrochemical impedance spectroscopy (EIS) in a 3.5 wt% NaCl solution at room temperature. EIS tests were performed on a CS165 electrochemical analyzer (Wuhan CorrTest Instruments Corp., Ltd., Wuhan, China) with a three-electrode system in a conventional three-electrode cell consisting of a large platinum sheet (2 × 2 cm$^2$), a saturated Ag/AgCl electrode and a steel electrode. The frequency range was $10^5$–$10^{-2}$ Hz at an open circuit potential with a disturbance amplitude of 5 mV.

## 3. Results and Discussion

### 3.1. Gelation Time of GPTMS-TEOS Hybrid Coatings

The time that the gel did not flow after inverting the vial was recorded as the gelation time. The GPTMS-TEOS (without DETA addition) hybrid exhibited long gelation times (over 100 days). As shown in Figure 1, increasing the amount of DETA added to the hybrids resulted in a significantly shorter gelation time. Furthermore, gelation was faster if the sol was prepared in large quantities (over 200 mL), as it was accompanied by an increase in temperature due to the release of heat. The enthalpy of reaction for the epoxy–amine reaction is known to be 100–118 kJ per equivalent epoxide [23], whereas the maximum heat of reaction for the hydrolysis of TEOS is much lower (12.9 kJ·mol$^{-1}$) [24]. Therefore,

the exothermic reaction between the epoxy of GPTMS and amine groups of DETA could accelerate the formation of the GPTMS-TEOS hybrid.

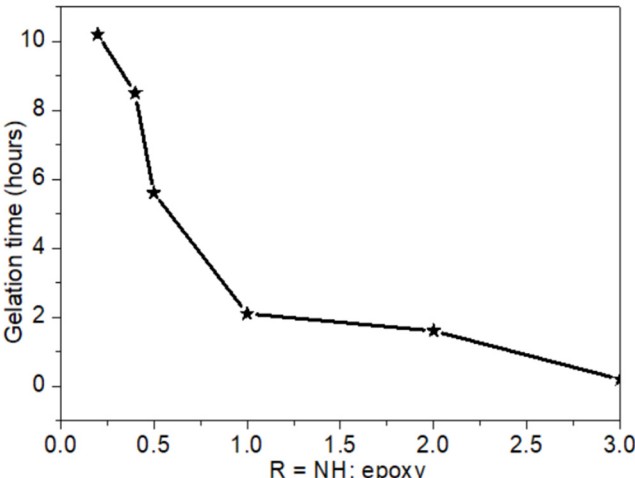

**Figure 1.** Gelation times for GPTMS-TEOS.

### 3.2. Structural and Morphological Characteristics of GPTMS-TEOS Hybrid Coatings

The X-ray diffraction patterns of the composites with different DETA additions are shown in Figure 2A. A broad diffraction peak near $2\theta = 23°$ was observed, which was formed by silica diffraction and is consistent with the literature description [25]. The presence of the broad diffraction peak indicates a disordered structure, suggesting that GPTMS forms amorphous $SiO_2$ via hydrolysis and condensation with TEOS under the catalytic action of DETA. The positions of the diffraction peaks were almost identical with increasing amounts of DETA addition, indicating that the addition of DETA had no effect on the formation of the final product of the GPTMS-TEOS composites. To further investigate the structure of the hybrid coatings, the functional groups of the composite were characterized via FT-IR, and the results are shown in Figure 2B. The characteristic peaks at 3414, 2919 and 2858 $cm^{-1}$ correspond to the stretching vibrations of -OH and -$CH_3$ bonds, respectively. The characteristic peak of the epoxy ring at 1250 $cm^{-1}$ indicates that not all the epoxy groups in GPTMS reacted completely. It is worth noting that the characteristic peaks at 1116, 1039 and 764 $cm^{-1}$ correspond to the stretching vibrations of Si-O-C, Si-O-Si, and Si-O bonds, respectively [26,27]. Combined with the previous XRD analysis, the result demonstrates that $SiO_2$ was in an amorphous state in the final product of the GPTMS-TEOS hybrid coating.

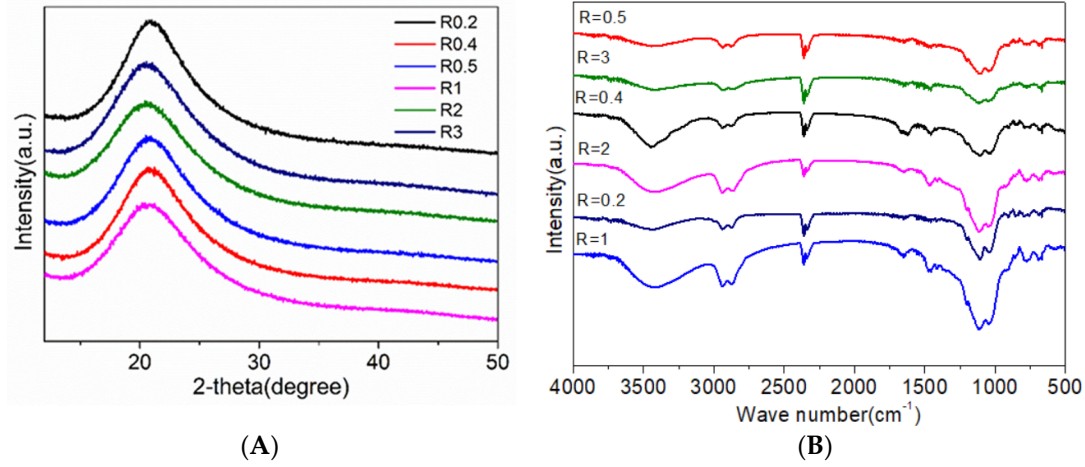

(**A**)          (**B**)

**Figure 2.** XRD patterns (**A**) and FT-IR spectra (**B**) of the GPTMS-TEOS composites.

The results of the above structural characterization confirmed the presence of silica in the final product. To further demonstrate the existence of silica in the composites, the surface morphology of the GPTMS-TEOS composites was characterized via SEM and TEM, during which the samples for the SEM test were produced by applying the hybrid coating to aluminum foil and drying, and tested samples for TEM were prepared by dispersing the GPTMS-TEOS composites in ethanol. It can be seen from Figure 3A that the surface of the coating is relatively dense with no obvious holes, which provides a physical barrier to the metal protection, reduces the penetration of electrolytes and slows down the corrosion rate. Figure 3B shows that there are spheres in the composite, which is consistent with the reported spherical structure of silica [5]. This further confirms the presence of silica production in the composite.

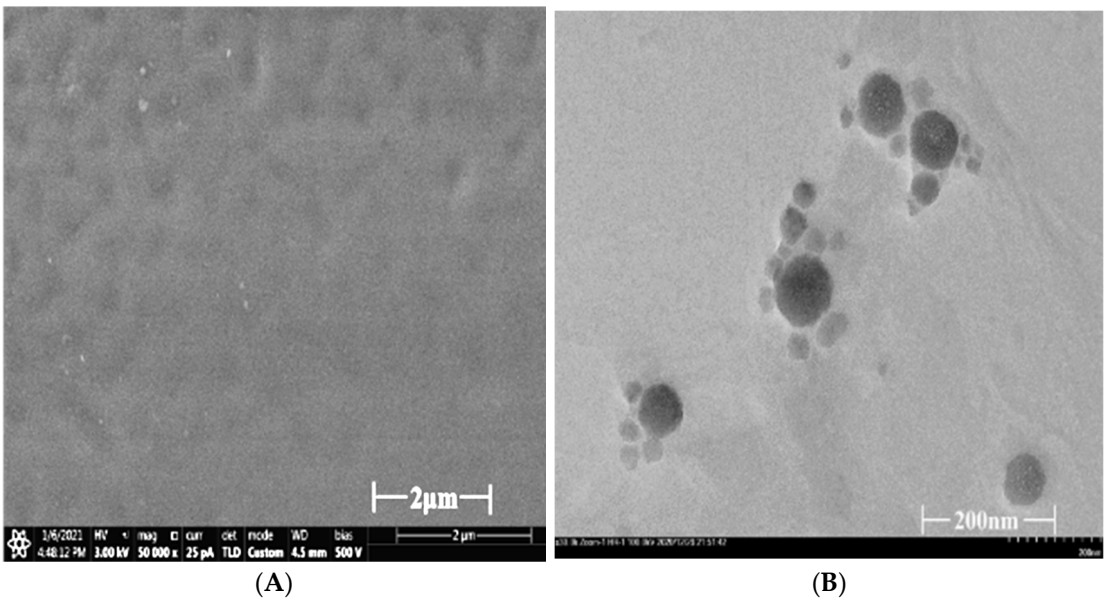

**(A)**　　　　　　　　　　　　　　　　　　　　　　　　　**(B)**

**Figure 3.** SEM images of coating (**A**) and TEM images of GPTMS-TEOS composites (**B**).

### 3.3. Thermal Stability of GPTMS-TEOS Hybrid Coatings

Thermal stability is an important property of coatings which affects the processibility and lifetime of the coatings [28,29]. The thermal decomposition behavior of the GPTMS-TEOS hybrid coating was analyzed using TGA, and the results are presented in Figure 4. Three processes exist for the degradation of all samples. The first step is in the low temperature range of 30–250 °C, which is associated with the evaporation of some residual small molecules (e.g., water, ethanol, and unreacted silanol); the second step is in the intermediate temperature range of 220–500 °C, which may be attributed to the decomposition of the glycidoxypropyl organic moieties from GPTMS; the last step is between 500–700 °C, which is associated with the degradation of the breaking of C-N bonds. The greater weight loss of R1, R2 and R3 before 250 °C is attributed to the higher degree of hydrolysis of the hybrid coating producing more evaporation of water. Between 400 and 600 °C, R1, R2 and R3 have better thermal stability, mainly because the more hydrolyzed hybrid coatings produce more silanol ($SiO_x(OH)_y$), which increases the heat resistance of the coating by forming $SiO_2$ at high temperatures [30]. Above 600 °C, the lower residual mass of R1, R2 and R3 is due to the fact that the more DETA in the hybrid coating produces more organic chains and the continued decomposition of the organic chains at higher temperatures leads to a continued loss of mass. In general, the high thermal stability exhibited by the hybrid coatings is attributable to the highly dense cross-linked hybrid structure provided by the curing agent (DETA) and the coupling agent (GPTMS).

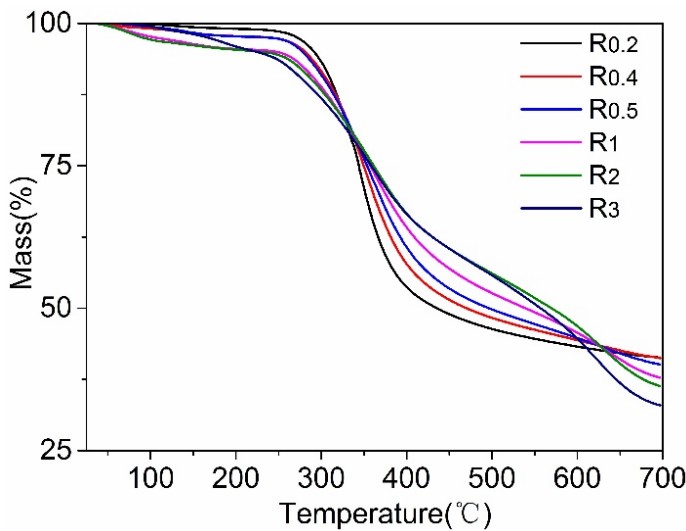

**Figure 4.** TGA curves of GPTMS-TEOS hybrid coating.

### 3.4. Anti-Corrosion Performance of GPTMS-TEOS Hybrid Coatings

EIS measurements were performed to characterize the corrosion resistance of GPTMS-TEOS hybrid coatings on the Q215 steel surface [31–33]. The electrochemical corrosion tests of Q215 steel, R0.2-coated Q215 steel (R0.2), R0.4-coated Q215 steel (R0.4), R0.5-coated Q215 steel (R0.5), R1-coated Q215 steel (R1), R2-coated Q215 steel (R2) and R3-coated Q215 steel (R3) were put in a 3.5 wt% NaCl solution in order to evaluate the anticorrosion performance of various hybrid coatings on Q215 steel.

Figure 5 shows the EIS results for Q215 steel and the various GPTMS-TEOS hybrid coatings after immersion for 1 w in the 3.5 wt% NaCl solution. The Nyquist diagram of the Q215 steel shown in Figure 5A presents only a capacitance loop, indicating that the reaction on the surface of Q215 steel is controlled by the charge transfer process [34]. It is well-known that the radius of the capacitance loop in the Nyquist diagram represents the polarization resistance of the working electrode, so a larger radius of the capacitance loop in the same corrosion system means better corrosion resistance [35,36]. The radius of the capacitance loop in the Nyquist plots of all the hybrid coatings shown in Figure 5B (especially in the inset graph in Figure 5B) are greater than that of Q215 steel, in descending order of R1 > R2 > R0.2 > R0.4 > R0.5 > R3 > Q215 steel, indicating that all the prepared hybrid coatings provide corrosion protection for Q215 steel. The results of the Nyquist plots show that GPTMS-TEOS hybrid coatings prepared with small or moderate amounts of DETA show a high resistance to the penetration of water and chloride ions. Although R1 and R2 have comparable barrier properties, delamination occurs more rapidly with R2. The rapid formation of siloxane networks that can occur at high levels of DETA has two consequences. Firstly, it reduces the concentration of silanol groups that can react with the hydroxyl on the surface of Q215 steel, and secondly, it gels too quickly, resulting in a hybrid coating without enough time to attach to the surface of Q215 steel. This conclusion is supported by the study by Rouw et al. [37], who emphasized the importance of the rheology of epoxy coatings on steel during the curing process. On the other hand, the geometrical constraints created by the rapid reaction between the DETA and epoxy groups may also be detrimental to adhesive bonding. In fact, this reaction creates additional binding sites in the inorganic network, which can hinder the ability of the film to form interfacial bonds with the metal surface.

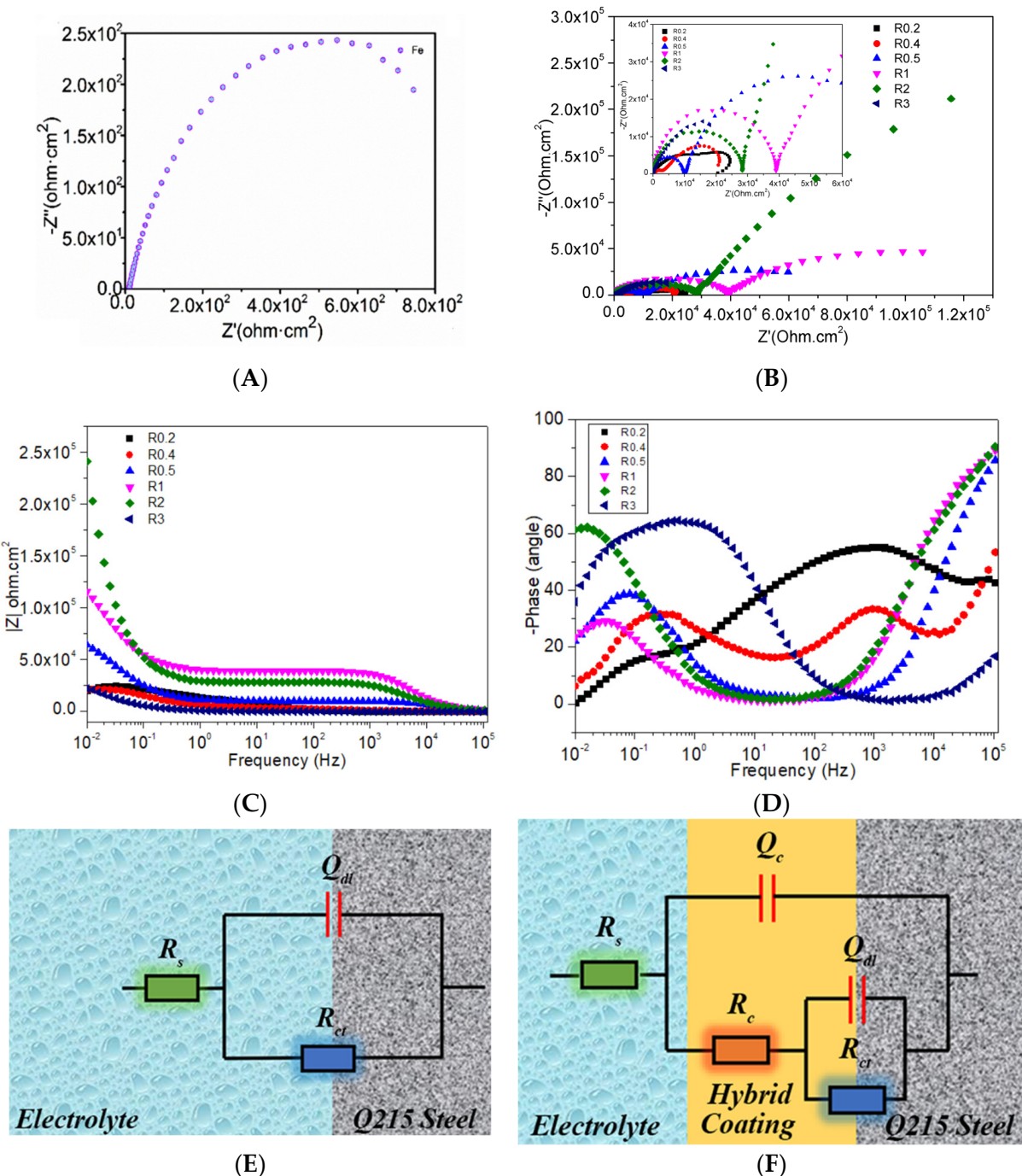

**Figure 5.** (**A**) Nyquist plot of Q215 steel, (**B**) Nyquist plot of hybrid coatings, (**C**) Bode plots of |Z| versus frequency, (**D**) Bode plots of phase angle versus frequency, and fitting circuits (**E,F**) applied to the EIS curves for Q215 steel and hybrid coatings immersed in 3.5 wt.% NaCl solution for 1 w, respectively.

　　　Figure 5C shows the Bode plots of the impedance modulus, |Z|, versus frequency. The impedance modulus at 0.01 Hz (|Z|0.01 Hz) can be used to evaluate the corrosion protection properties of the coatings. The |Z|0.01 Hz values for steel samples coated with GPTMS-TEOS hybrid coatings are roughly in the range of $10^3 \sim 10^5$ Ω·cm$^2$, exceeding the |Z|0.01 Hz values for Q215 steel samples by one to three orders of magnitude. It is noteworthy that the |Z|0.01 Hz values tend to increase and then decrease as the DETA content of the hybrid coatings increases, indicating that the addition of the right amount of DETA has a positive effect on the hybrid coatings. However, the high DETA content results

in a low crosslinking of the siloxane network. Additionally, the unreacted amino groups in the hybrid coating are hydrophilic, and both these groups contribute to the absorption of moisture by the coating and reduce the corrosion protection properties, so the |Z|0.01 Hz value of R3 decreases significantly. Figure 5D shows the Bode plots of the phase angle versus frequency. The high-frequency capacitive arc corresponds to the barrier properties of the GPTMS-TEOS hybrid coatings, and the broad arc observed at low frequencies is related to the charge transfer process.

In order to further analyze the corrosion resistance of the hybrid coatings on Q215 steel, the above EIS curves were fitted using the fitting circuit, as shown in Figure 5E,F. The results are presented in Table 2. In these circuits, $R_s$ is the solution resistance, $Q_c$ is the constant phase element of the coatings, $R_c$ is the coating resistance, $Q_{dl}$ is double-layer capacitance, and $R_{ct}$ is the charge transfer resistance [38–40]. Generally, lower $Q_c$ and $Q_{dl}$ values illustrated the less corrosive media's penetration into the hybrid coatings, while higher $R_c$ and $R_{ct}$ values represent the better shielding performance of the hybrid coatings [36]. As the DETA content increases, the $Q_c$ and $Q_{dl}$ values of the hybrid coating both show a trend of decreasing and then increasing, with R1 showing lower $Q_c$ and $Q_{dl}$ values, indicating a stronger ability to inhibit electrolyte penetration. In contrast, the $R_c$ and $R_{ct}$ values of the hybrid coatings show a pattern of increasing and then decreasing as the DETA content increases, with R1 having the highest $R_c$ and $R_{ct}$ values and showing the best shielding performance.

**Table 2.** Electrochemical parameters obtained from simulation of EIS results.

| Samples | $R_s$ ($\Omega \cdot cm^2$) | $Q_c$ | | $R_c$ ($\Omega \cdot cm^2$) | $Q_{dl}$ | | $R_{ct}$ ($\Omega \cdot cm^2$) |
|---|---|---|---|---|---|---|---|
| | | $Y_0$ ($F \cdot cm^{-2}$) | $N$ | | $Y_0$ ($F \cdot cm^{-2}$) | $n$ | |
| Q215 | 6.5 | / | / | / | $6.48 \times 10^{-4}$ | 0.782 | 386 |
| R$_{0.2}$ | 1.74 | $7.82 \times 10^{-6}$ | 0.619 | $1.783 \times 10^4$ | $9.62 \times 10^{-5}$ | 0.828 | $8.63 \times 10^3$ |
| R$_{0.4}$ | 1.65 | $2.39 \times 10^{-5}$ | 0.371 | $6.46 \times 10^3$ | $4.84 \times 10^{-5}$ | 0.851 | $2.31 \times 10^3$ |
| R$_{0.5}$ | 2.12 | $2.16 \times 10^{-9}$ | 0.800 | $9.94 \times 10^3$ | $6.93 \times 10^{-5}$ | 0.800 | $7.61 \times 10^4$ |
| R$_1$ | 2.03 | $1.79 \times 10^{-9}$ | 0.954 | $3.86 \times 10^4$ | $5.56 \times 10^{-5}$ | 0.763 | $4.56 \times 10^6$ |
| R$_2$ | 3.12 | $3.93 \times 10^{-9}$ | 0.915 | $2.79 \times 10^4$ | $3.97 \times 10^{-5}$ | 0.813 | $1.49 \times 10^6$ |
| R$_3$ | 7.41 | $5.84 \times 10^{-5}$ | 0.383 | $1.79 \times 10^4$ | $1.78 \times 10^{-4}$ | 0.844 | $9.64 \times 10^4$ |

In summary, the addition of DETA catalyzes the hydrolysis reaction, but it greatly increases the rate of the self-condensation reaction. GPTMS is more reactive to self-condensation reactions than TEOS is. The excess of active hydrogen in the amine group of DETA accelerates the kinetic reaction of the system and shortens the time until gelation formation, but at the same time this leads to a reduction in the integrity of the inorganic network in the final dry film. This leads to a rapid deterioration of the coating properties. At a high amine content, the rapid condensation of silanol groups forms Si-O-Si bridge bonds. As a result, there are virtually no silanol groups to react with the hydroxyl groups on the surface of the Q215 steel substrate to form Fe-O-Si bonds. This results in a hybrid coating with no solid reaction point on the metal surface and rapid delamination upon immersion in a NaCl solution, reducing its corrosion protection capability. The GPTMS-TEOS hybrid coating obtained with the appropriate amount of DETA has the best corrosion protection properties because the complete chemical reaction of the amino and epoxy groups results in a dense protective film, which is at this point the most hydrophobic and hardest, and therefore has the best protection properties in the corrosive medium of NaCl.

### 3.5. Anticorrosion Mechanism of GPTMS-TEOS Hybrid Coatings

During preparation, GPTMS and TEOS were added to ethanol, and some water was added. The mixture was stirred magnetically for 4 h at room temperature to prepare the silanol. DETA was added to the above-hydrolyzed silane–sol. We can see that the large spheres of silica were first connected by GPTMS and then the epoxy groups of the GPTMS were linked by the DETA.

As is shown in Figure 6, the GPTMS-TEOS hybrid coatings with moderate amounts of DETA exhibit excellent anti-corrosion performance on a Q215 steel surface due to the following factors. Firstly, the appropriate amount of DETA acts as both a curing agent and a catalyst for the GPTMS-TEOS hybrid coating, forming a dense protective film on the surface of Q215 steel to slow down the penetration of corrosive media [41]; secondly, the amorphous $SiO_2$ formed in the GPTMS-TEOS hybrid coating greatly prolongs the diffusion channel of corrosive media [42,43]; finally, the silicon hydroxyl group in the hybrid coating can bond with the hydroxyl group on the surface of Q215 steel to increase the adhesion of the hybrid coating on Q215 steel [44].

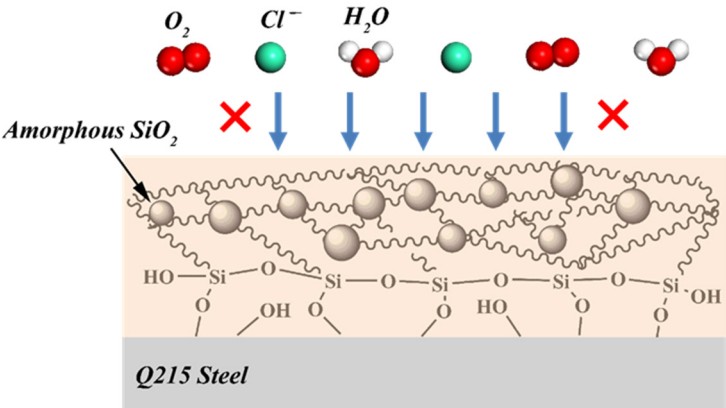

**Figure 6.** Scheme of anticorrosion mechanism of GPTMS-TEOS hybrid coatings on Q215 steel surface.

### 3.6. Salt Spray Test of GPTMS-TEOS Hybrid Coatings

Electrochemical corrosion tests show that when the molar ratio of ethyl silicate to the silane coupling agent is 0.5 (T0.5), the coating has the highest impedance value, and the impedance value remains at 109 $\Omega$ cm$^2$ after immersion in salt water for 96 h. To further investigate whether or not the coating has long-term stability, the sample was immersed in a 5 wt.% NaCl solution for a salt spray test. The salt spray test image is shown in Figure 7. As can be seen from the image, when the steel plate is immersed in salt water for 4 days, there is no rusting phenomenon on the surface of the steel plate except for water droplets. When the time increases to 44 days, the metal begins to rust. When the immersion time increases to 68 days, obvious rusting appears on the metal substrate surface, but the metal is not completely corroded. When corrosion occurs, chloride ions penetrate the surface of the substrate, thus accelerating the corrosion of the metal with the increasing immersion time. The results above demonstrate that an organic–inorganic hybrid coating prepared with appropriate amounts of ethyl silicate and epoxy resin can improve the excellent corrosion protection efficiency. Because the added TEOS can be hydrolyzed into a large number of Si-OH groups, Si-OH groups can condense to form a Si-O-Si mesh structure, and the formation of a hydrophobic Si-O-Si network improves the barrier effect of the coating and greatly improves the corrosion resistance of the coating.

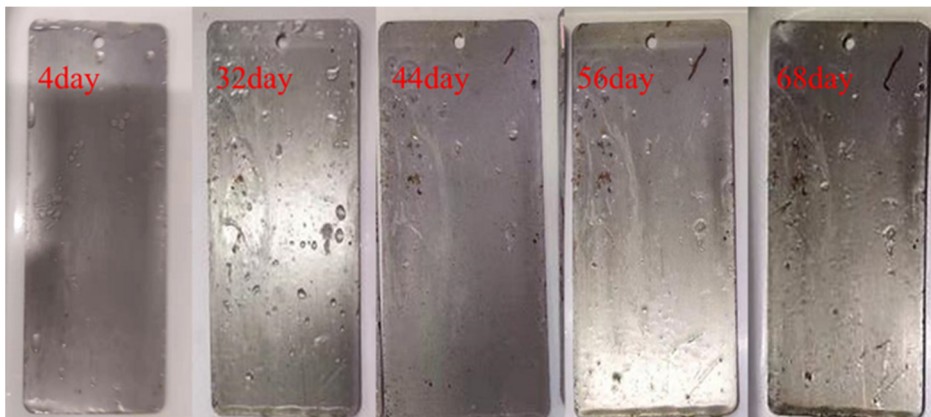

**Figure 7.** Images of salt spray test.

## 4. Conclusions

GPTMS-TEOS hybrid coatings were successfully prepared at room temperature using GPTMS and TEOS as reaction raw materials and DETA as a curing agent and catalyst. To determine the effect of the content of DETA on the gelation time of the coating, the thermodynamic and corrosion resistance of the coating was investigated systematically. The gelation time was significantly reduced by the increased content of DETA, which is related to the heat generated by the condensation reaction. The prepared GPTMS-TEOS hybrid coatings contain amorphous $SiO_2$ as demonstrated via XRD, FTIR, SEM, and TEM characterization. The amount of DETA influences the thermal stability of the coating by affecting the production of amorphous silica in the hybrid coating. The better thermal stability of the coatings under amine-rich conditions between 500~600 °C is attributed to the breaking of C-N bonds. Based on the EIS data, it was concluded that when the ratio of the active amine to the epoxy value was 1, the prepared hybrid coating had optimum corrosion protection, with an impedance value three orders of magnitude higher than that of bare steel. This study provides an informative guideline for the preparation of high-performance organic–inorganic hybrid coatings and presents a deep study on the corrosion protection mechanism of hybrid coatings.

The prepared coating is a little bit brittle compared with the traditional epoxy coatings. In a future investigation, we should improve the strength of the hybrid coating.

**Author Contributions:** Conceptualization, R.H. and Z.J.; methodology, S.Y. And Z.J.; validation, S.Y., Z.J. and R.H.; investigation, J.X. and R.H.; resources, R.H.; data curation, S.Y.; writing—original draft preparation, Z.J. and S.Y.; writing—review and editing, S.Y. and R.H.; supervision, R.H.; project administration, R.H.; funding acquisition, R.H. All authors have read and agreed to the published version of the manuscript.

**Funding:** This research was financially supported by the National Natural Science Foundation of China (NSFC, no. 22278080), Minjiang Scholarship of Fujian Province (no. Min-Gaojiao [2010]-117), Central Government-Guided Fund for Local Economic Development (no. 830170778), R&D Fund for Strategic Emerging Industry of Fujian Province (no. 82918001), and Analytical Testing Fund of Qingyuan Innovation Laboratory of Fujian Province.

**Institutional Review Board Statement:** Not applicable.

**Informed Consent Statement:** Not applicable.

**Data Availability Statement:** Not applicable.

**Conflicts of Interest:** The authors declare no conflict of interest.

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
