# Peer review of "Influence of DETA on Thermal and Corrosion Protection Properties of GPTMS-TEOS Hybrid Coatings on Q215 Steel"

_coatings, doi:10.3390/coatings13071145_

Round 1

Reviewer 1 Report

Find below comments which must be considered to improve the quality of the presented paper:

Line 14: "which characterized " correct to "was characterized".

Lines 12, 44, 89: "ethyl silicate (TEOS)", better define as tetraethylorthosilicate (TEOS)

Line 35: define "GPD" in the text.

Lines 64 – 67: About the formation and structure at molecular level of the GPTMS-TEOS system which is an important discussion related to the material studied in this work, the authors can improve the introduction and cite important references such as:

D. R. Vollet et al. , Rod-like particles growing in sol–gel processing of 1:1 molar mixtures of 3-glycidoxypropyl­tri­meth­oxy­silane and tetra­eth­oxy­silane, J. Appl. Cryst. (2017). 50489-497.   https://doi.org/10.1107/S1600576717002357.

Plinio Innocenzi et al., Sol-gel reactions of 3-glycidoxypropyltrimethoxysilane in a highly basic aqueous solution, Dalton Trans., (2009), 9146–9152 DOI: 10.1039/b905830c.

M.R. Gizdavic-Nikolaidi et al., Structure and properties of GPTMS/DETA and GPTMS/EDA hybrid polymers, Current Applied Physics 8 (2008) 300–303. https://doi.org/10.1016/j.cap.2007.10.019.

Line 79: “investigate their macroscopic properties” by scanning electron microscopy (SEM), transmission electron microscopy (TEM). These techniques are used to investigate “microscopic” properties, NOT macroscopic.

Line 125: “The time that the gelation does not flow”, correct to: “The time that the gel does not flow”

Line 170: The formatting of Figure 4 need be improved.

Line 175 – The atmosphere used for TGA analysis must be informed.

Line 178 – Discussion about the TGA results (Fig. 4) needs profound revision. The second step of mass loss (220-500 oC) is due not only to dehydration of silanol, but also related to decomposition of the glycidoxypropyl organic moieties from GPTMS.

Line 199: The graphs A-D  (Figure 5) should have the same size. The powers of ten (eg: 105) in y axes can be arranged in the legend of Z and removed from the y scale, improving the visual clarity of these graphs. The inset in graph B has very poor image quality.

Line 200 – Legend of Fig. 5 do not mention the inset in graph B.

Line 212 –  The radius of the capacitance loop in the Nyquist plots shown in Fig. 5B in descending order of R1 > R2 > R0.2 > R0.4 > R0.5 > R3 > Q215 steel, is not clear looking to the graphs. It is not possible from Fig. 5B to identify this radius sequence (R1 > R2 > R0.2 > R0.4 > R0.5 > R3 > Q215) as stated in the text.

Line 272 – In the scheme presented (Fig. 6), authors must inform in the legend “who are” the large spheres designed in the amorphous SiO2 region. And also inform who are the moieties that connect these large spheres.

See refs:

D. A. Monteiro et al. Proton conduction mechanisms in GPTMS/TEOS-derived organic/silica hybrid films prepared by sol-gel process, Synthetic Metals 267, September 2020, 116448. https://doi.org/10.1016/j.synthmet.2020.116448

D. R. Vollet et al. , Rod-like particles growing in sol–gel processing of 1:1 molar mixtures of 3-glycidoxypropyl­tri­meth­oxy­silane and tetra­eth­oxy­silane, J. Appl. Cryst. (2017). 50489-497.https://doi.org/10.1107/S1600576717002357.

 Line 288: "sistemerycally", correct to "systematically".

line 14: "which characterized " correct to "was characterized"

line 35: define "GPD".

Line 288: "sistemerycally", correct to "systematically".

Author Response

Line 14: "which characterized " was changed to "was characterized".

Response: Thank you for giving the kind suggestions. Under your guidance, the error was corrected.

Lines 12, 44, 89: "ethyl silicate (TEOS)", better define as tetraethylorthosilicate (TEOS)

Response: Under your suggestion, TEOS was defined as “tetraethylorthosilicate (TEOS)”.

Line 35: define "GPD" in the text.

Response: Thank you for giving the kind suggestions. The "GDP" was defined as "Gross Domestic Product" in the revised manuscript.

Lines 64 – 67: About the formation and structure at molecular level of the GPTMS-TEOS system which is an important discussion related to the material studied in this work, the authors can improve the introduction and cite important references such as:

  1. R. Vollet et al.Rod-like particles growing in sol–gel processing of 1:1 molar mixtures of 3-glycidoxypropyl­tri­meth­oxy­silane and tetra­eth­oxy­silaneJ. Appl. Cryst.(2017). 50, 489-497.   https://doi.org/10.1107/S1600576717002357.

Plinio Innocenzi et al.Sol-gel reactions of 3-glycidoxypropyltrimethoxysilane in a highly basic aqueous solution, Dalton Trans., (2009), 9146–9152 DOI: 10.1039/b905830c.

  1. Gizdavic-Nikolaidi et al., Structure and properties of GPTMS/DETA and GPTMS/EDA hybrid polymers, Current Applied Physics 8(2008) 300–303. https://doi.org/10.1016/j.cap.2007.10.019.

Response: The above papers have been cited in the revised manuscript (Refs. 17, 18, 19).

Line 79: “investigate their macroscopic properties” by scanning electron microscopy (SEM), transmission electron microscopy (TEM). These techniques are used to investigate “microscopic” properties, NOT macroscopic.

Response: Thank you for giving us your kind suggestions. "investigate their macroscopic properties" was corrected as "investigate their microscopic properties" in the revised manuscript.

Line 125: “The time that the gelation does not flow”, correct to: “The time that the gel does not flow”

Response: Thank you for giving the kind suggestions. "The time that the gelation does not flow" was corrected to "The time that the gel does not flow" in revised manuscript.

Line 170: The formatting of Figure 4 need be improved.

Line 175 – The atmosphere used for TGA analysis must be informed.

Response: Thank you for giving us your kind suggestions. The atmosphere used for TGA analysis (N2 atmosphere) was informed in revised manuscript.

Line 178 – Discussion about the TGA results (Fig. 4) needs profound revision. The second step of mass loss (220-500 oC) is due not only to dehydration of silanol, but also related to decomposition of the glycidoxypropyl organic moieties from GPTMS.

Response: Thank you for giving the kind suggestions. The discussion about the “the TGA results” (Fig. 4)  has been corrected in the revised manuscript.

Line 199: The graphs A-D  (Figure 5) should have the same size. The powers of ten (eg: 105) in y axes can be arranged in the legend of Z and removed from the y scale, improving the visual clarity of these graphs. The inset in graph B has very poor image quality.

Line 200 – Legend of Fig. 5 do not mention the inset in graph B.

Response: Thank you for giving the kind suggestions. The inset graph in Fig. 5B has been mentioned in the revised manuscript.

Line 212 –  The radius of the capacitance loop in the Nyquist plots shown in Fig. 5B in descending order of R1 > R2 > R0.2 > R0.4 > R0.5 > R3 > Q215 steel, is not clear looking to the graphs. It is not possible from Fig. 5B to identify this radius sequence (R1 > R2 > R0.2 > R0.4 > R0.5 > R3 > Q215) as stated in the text.

Line 272 – In the scheme presented (Fig. 6), authors must inform in the legend “who are” the large spheres designed in the amorphous SiO2 region. And also inform who are the moieties that connect these large spheres.

See refs:

  1. A. Monteiro et al. Proton conduction mechanisms in GPTMS/TEOS-derived organic/silica hybrid films prepared by sol-gel process, Synthetic Metals 267, September 2020, 116448. https://doi.org/10.1016/j.synthmet.2020.116448
  2. R. Vollet et al.Rod-like particles growing in sol–gel processing of 1:1 molar mixtures of 3-glycidoxypropyl­tri­meth­oxy­silane and tetra­eth­oxy­silaneJ. Appl. Cryst. (2017). 50, 489-497. https://doi.org/10.1107/S1600576717002357.

Line 288: "sistemerycally", correct to "systematically".

Response: Thank you for giving the kind suggestions. The amorphous silica was pointed out in Fig. 6 of the revised manuscript. "sistemerycally" has been corrected  to "systematically" in revised manuscript.

Reviewer 2 Report

Title: Study on Influence of DETA on Thermal and Corrosion Protection Properties of GPTMS-TEOS Hybrid Coatings over Q215  Steel

In this study, the authors reported GPTMS-TEOS hybrid coatings for corrosion protection of Q215 steel using diethy lenetriamine (DETA) as curing agent.

There are so many lapses in the manuscript especially corrosion part and needs major revision.

1. Introduction is not enough. Authors need to introduce more literature having such coatings for corrosion protection of steel.

2. Authors mentioned that in the Nyquist diagram only a capacitance loop is there but in figures there are two loops.

3. For corrosion protection study, polarization test should be performed, only EIS results are not sufficient.

Minor improvement is required

Author Response

  1. Introduction is not enough. Authors need to introduce more literature having such coatings for corrosion protection of steel.

Response: Three more papers were cited in the revised manuscript (Refs. 16, 17, 18).

  1. Authors mentioned that in the Nyquist diagram only a capacitance loop is there but in figures there are two loops.

Response: The two figures in this paper are for the performance of Q215 steel substrate under different conditions. Figure 5 A is a Nyquist diagram without anti-corrosion materials; while Figure 5 B is the Nyquist diagram with anti-corrosion material

  1. For corrosion protection study, polarization test should be performed, only EIS results are not sufficient.

Response: Thanks for the reminder, your comments are very meaningful to us. In the paper, we performed the EIS test, and also carried out a salt spray test. The good protection effect was observed, and hence the polarization test was not presented in the manuscript.

Reviewer 3 Report

The paper was massively copied from other works, such as F.X. Perrin, F. Ziarelli, A. Dupuis. "Relation between the corrosion resistance and the chemical structure of hybrid sol-gel coatings with interlinked inorganic-organic network" , Progress in Organic Coatings, 2020; Zhenzhen Jia, Ruoyu Hong. "Anticorrosive and photocatalytic properties research of epoxysilica organic–inorganic coating" , Colloids and Surfaces A: Physicochemical and Engineering Aspects, 2021; Huaiyin Chen, Huizhou Fan, Nan Su, Ruoyu Hong, Xuesong Lu. "Highly hydrophobic polyaniline nanoparticles for anti-corrosion epoxy coatings" , Chemical Engineering Journal, 2021.

The structure of the paper is not mentioned in the first chapter.

The title of chapter 2 is called Experimental. Experimental what? Methods?

There is no reference to table 1 in the text.

The title of chapter 4 is the very last line of page 8, but its content is found on the following page. Please place them together.

The conclusions section must mention future work.

The paper was massively copied from other works, such as F.X. Perrin, F. Ziarelli, A. Dupuis. "Relation between the corrosion resistance and the chemical structure of hybrid sol-gel coatings with interlinked inorganic-organic network" , Progress in Organic Coatings, 2020; Zhenzhen Jia, Ruoyu Hong. "Anticorrosive and photocatalytic properties research of epoxysilica organic–inorganic coating" , Colloids and Surfaces A: Physicochemical and Engineering Aspects, 2021; Huaiyin Chen, Huizhou Fan, Nan Su, Ruoyu Hong, Xuesong Lu. "Highly hydrophobic polyaniline nanoparticles for anti-corrosion epoxy coatings" , Chemical Engineering Journal, 2021.

The structure of the paper is not mentioned in the first chapter.

The title of chapter 2 is called Experimental. Experimental what? Methods?

There is no reference to table 1 in the text.

The title of chapter 4 is the very last line of page 8, but its content is found on the following page. Please place them together.

The conclusions section must mention future work.

Author Response

1The paper was massively copied from other works, such as F.X. Perrin, F. Ziarelli, A. Dupuis. "Relation between the corrosion resistance and the chemical structure of hybrid sol-gel coatings with interlinked inorganic-organic network" , Progress in Organic Coatings, 2020; Zhenzhen Jia, Ruoyu Hong. "Anticorrosive and photocatalytic properties research of epoxysilica organic–inorganic coating" , Colloids and Surfaces A: Physicochemical and Engineering Aspects, 2021; Huaiyin Chen, Huizhou Fan, Nan Su, Ruoyu Hong, Xuesong Lu. "Highly hydrophobic polyaniline nanoparticles for anti-corrosion epoxy coatings" , Chemical Engineering Journal, 2021.

Response:

In the three articles mentioned above, "Relation between the corrosion resistance and the chemical structure of hybrid sol-gel coatings with interlinked inorganic-organic network", the article uses triethylenetetramine (TETA) as a material for sol formulation. The material used in this article is diethylenetriamine (DETA) as raw material, the materials used in the two are different, and the corresponding process is also biased, in order to test the coating performance during the test process, the test process is a conventional test process, and there is no plagiarism.

Anticorrosive and photocatalytic properties research of epoxysilica organic–inorganic coating, and Highly hydrophobic polyaniline nanoparticles for anti-corrosion epoxy coatings, The two papers were published by the project team, and there was no plagiarism, which was the result of different stages in the process of the project.

2The structure of the paper is not mentioned in the first chapter.

Response:

A description of the structure of the paper is added at the end of the first part of the paper;

3The title of chapter 2 is called Experimental. Experimental what?

Response:

Thank you for the review's suggestions, Chapter 2 is the preparation process of new coating materials, and the title of Chapter 2 is modified as “2. Preparation of hybrid coatings”

4There is no reference to table 1 in the text.

Response:

Thanks to the reviewers' suggestions, we revised the content.

5、、The title of chapter 4 is the very last line of page 8, but its content is found on the following page. Please place them together.

Response:

感谢审稿人的建议,我们修改了内容。

6结论部分必须提及未来工作。

本文采用的涂层制备方法合理,本文实验中制备的涂层也存在脆性的缺点,后期涂层制备工艺中加入双酚苯三环,发表在《有机涂料进展》,2020;贾珍珍, 洪若宇.“环氧二氧化硅有机-无机涂料的防腐和光催化性能研究”论文;本文总结了本文的初步研究过程和方法。

Reviewer 4 Report

The manuscript reports on the structural, and electrochemical properties of anti-corrosion silica-epoxy coatings, based on TEOS, GPTMS, and DETA precursors, to protect Q215 steel. The authors present some interesting aspects regarding hybrid preparation and thermal properties upon increasing the concentration of the DETA curing agent. However, the manuscript contains several affirmations regarding silica clusters, adhesion, hardness, and hydrophobicity without providing experimental evidence. Also, the proposed structural model for the R1 sample is not supported by structural data, here NMR study would be useful. The poor EIS performance obtained for a very short exposure period (1 h), is close to that of bare steel and far below of that reported in studies cited in the Introduction. Therefore, the affirmation that “moderate amounts of DETA exhibit excellent anti-corrosion performance on Q215 steel” is not sustainable, making the practical use of this coating system questionable. The low corrosion resistance is probably a consequence of a heterogeneous film structure containing large silica agglomerates (Fig. 3). In view of these points and several other issues, listed below, the manuscript is not suitable for publication. Nevertheless, the following points might be addressed to improve the work.

Title:

1- Exclude “Study” from the title and substitute “over” with “on”.

Introduction:

2- Several important references of durable high-performance organic-inorganic coatings based on silica-PMMA hybrids on carbon steel are missing.

Experimental:

3- L. 98: Comment on the reason to use a GPTMS to TEOS ratio of 3.

4- 2.3: Inform the atmosphere of TGA measurements.

Results and discussion:

5- Fig. 2, L. 149: Comment on the incomplete reaction of epoxy groups at high DETA content.

6- L. 159: This statement is not clear, please comment: “The results of the above structural characterization confirmed the presence of silica in the final product.”

7- Fig. 3B: Identify the nature of spheres using EDS. Do the proportions of spheres scale with the DETA content?

8- Please provide information on the coating thickness (cross-section SEM).

9- L. 178: Reconsider the hybrid decomposition in the “intermediate temperature range of 220-500 °C, which may be attributed to dehydration of silanol;” In this temperature range a depolymerization reactions of the organic structure takes place. It is well known that dehydration of silanol occurs at T > 500 °C. 

10- Fig. 4 and Conclusions: The degradation event at 500-600 °C of amine-rich samples might be related to the breaking of C-N bonds rather than the “to greater production of SiO2.” 

11- L. 189: To support the argument of “…high thermal stability…” please include in a table as criterium for the thermal stability the temperature at the mass loss of 5% (10%).

12- L. 215: Considering a very short immersion period of 1 h, reconsider the affirmation that “…all the prepared hybrid coatings provide corrosion protection for Q215 steel.” and that “…small or moderate amounts of DETA show a high resistance to the penetration of water and chloride ions…”.

13- L. 222: Clarify the expression “…infiltrate the surface of Q215 steel.”

14- L. 264: Confirm the statement: “As a result, there are virtually no silanol groups to react with the hydroxyl groups on the surface of the Q215 steel substrate to form Fe-O-Si bonds.”, providing results of DETA content dependence of the adhesion strength to the steel substrate, using a standard ANSI method.

15- L. 270: To support the affirmation: “…which is at this point the most hydrophobic and hardest.”, provide hardness and contact angle measurement.

Minor issues:

- L. 45: Correct TEOS: tetraethyl orthosilicate

- Fig. 4: Correct the ticks of the Mass axis

- Fig. 5 B-C: Correct the sample sequence in the legend and include fitting curves.

 Moderate editing of the English language is necessary.

Author Response

  • Exclude “Study” from the title and substitute “over” with “on”.

Response: Thank you for giving the kind suggestions. The title has been corrected to  "Influence of DETA on Thermal and Corrosion Protection Properties of GPTMS-TEOS Hybrid Coatings on Q215 Steel".

Introduction:

2- Several important references of durable high-performance organic-inorganic coatings based on silica-PMMA hybrids on carbon steel are missing.

Response: Sorry to infrom you, the PMMA (methyl methacrylate) had nothing to do with our work. Our coating is an organic-inorganic nano hybrid coating formed by the oligomer obtained by prepolymerization of three raw materials and inorganic nano-silica generated in situ.

And according to the requirements of another reviewer, three more papers, Refs. 17, 18 and 19, were cited in the revised manuscript.

Experimental:

3- L. 98: Comment on the reason to use a GPTMS to TEOS ratio of 3.

4- 2.3: Inform the atmosphere of TGA measurements.

Response: Thank you for giving the kind suggestions. The best ratio of GPTMS to TEOS is 3:1 obtained by experimental investigation and we are still working hard to find the reason. The atmosphere used for TGA analysis (N2 atmosphere) was informed in revised manuscript.

Results and discussion:

  • 2, L. 149: Comment on the incomplete reaction of epoxy groups at high DETA content.

Response: During the preparation of the samples, we used 0.03 mol GPTMS and 0.01 mol TEOS, and different dosage of DETA. We found that the sample with the Rx = 0.5 is the best. Hence, there could be some unreacted epoxy groups.

  • 159: This statement is not clear, please comment: “The results of the above structural characterization confirmed the presence of silica in the final product.”

Response: In Sect. 3.2, the XRD patterns in Fig. 2 illustrated that there are silica in the nano composite coating, while there could be seen the presence of silica nano particles in Fig. 3. Hence, in the second paragraph we concluded the presence of silica in the final product.

  • 3B: Identify the nature of spheres using EDS. Do the proportions of spheres scale with the DETA content?

Response: In the first paragraph of Sec. 3.2, we concluded that GPTMS forms amorphous SiO2 by hydrolysis and condensation with TEOS under the catalytic action of DETA. The proportions of spheres scales with the TEOS content, but not with that of the DETA. The DETA severs as a catalyst in this case.

  • Please provide information on the coating thickness (cross-section SEM).

Response: It is a very good comment. We already added in the revised manuscript: The thickness of the coating prepared for the salt spray tests was about 80 nm.

  • 178: Reconsider the hybrid decomposition in the “intermediate temperature range of 220-500 °C, which may be attributed to dehydration of silanol;” In this temperature range a depolymerization reactions of the organic structure takes place. It is well known that dehydration of silanol occurs at T > 500 °C. 

Response: Yes, it is a very good comment. It is the depolymerization for the weight loss at the temerature range of 220 to 500 °C. We alreay modified the manuscript.

10- Fig. 4 and Conclusions: The degradation event at 500-600 °C of amine-rich samples might be related to the breaking of C-N bonds rather than the “to greater production of SiO2.” 

Response: Thank you for giving the kind suggestions. We have corrected the corresponding sentences in Sec. 3.3 of the revised manuscript.

  • 189: To support the argument of “…high thermal stability…” please include in a table as criterium for the thermal stability the temperature at the mass loss of 5% (10%).

Response: As shown in Table 1, the dosage of DETA was varied in the experiments. It was found that the best one is the R0.5, tested by the electrochemical measurement, as shown in Table 2.

For the thermal tests using the TGA for the sample prepared with the different ratios,

the results are demonstrated in Fig.4. Sorry for not listing the data using a table.

  • 215: Considering a very short immersion period of 1 h, reconsider the affirmation that “…all the prepared hybrid coatings provide corrosion protection for Q215 steel.” and that “…small or moderate amounts of DETA show a high resistance to the penetration of water and chloride ions…”.

Response: Yes, it is again a good comment. We made a mistake, it should be w (week) not h (hour). The mistake as corrected in the revised manuscript.

13- L. 222: Clarify the expression “…infiltrate the surface of Q215 steel.”

Response: We are sorry for the misleading. Under your guidance, the sentence has been corrected to "it gels too quickly, resulting in the hybrid coating without enough time to attach to the surface of Q215 steel".

  • 264: Confirm the statement: “As a result, there are virtually no silanol groups to react with the hydroxyl groups on the surface of the Q215 steel substrate to form Fe-O-Si bonds.”, providing results of DETA content dependence of the adhesion strength to the steel substrate, using a standard ANSI method.

Response: The GPTMS not the GPTMS might react with the hydroxyl groups on the surface of the Q215 steel substrate to form Fe-O-Si bonds at the room temperature. Since there is water during the preparation of the hybrid coating, as demonstrated in Sec. 2.2, there is no free GPTMS in the suspension. Hence, there will be not Fe-O-Si bonds on the steel substrate.

  • 270: To support the affirmation: “…which is at this point the most hydrophobic and hardest.”, provide hardness and contact angle measurement.

Response: We only performed the electrochemical measurement, and no hardness and contact angle measurement. We already deleted the sentence in the revised manuscript.

Minor issues:

16- L. 45: Correct TEOS: tetraethyl orthosilicate

Response: Thank you for giving the kind suggestion. TEOS has been corrected to “tetraethylorthosilicate (TEOS)”.

- Fig. 4: Correct the ticks of the Mass axis

Response: Thanks for the suggestion. In Fig.4, the ordinate axis is the percentage of mass, and the part close to the abscissa is not reflected in the experimental range, so the start of the ordinate is 25%, which is the valid part of the intercept.

- Fig. 5 B-C: Correct the sample sequence in the legend and include fitting curves.

Response: Thanks for the suggestion. Figs. 5 B and C were redrawn in the revised manuscript.

Reviewer 5 Report

Overall, the study on the influence of DETA on the thermal and corrosion protection properties of GPTMS-TEOS hybrid coatings over Q215 steel is interesting and provides valuable insights into the performance of these coatings. However, there are several areas that require major revision before this work can be accepted for publication.

1.     The abstract is too brief and lacks important details. It would be helpful to provide more context and background information about the study, as well as a brief summary of the main findings and implications.

2.     The introduction should be expanded to provide more background information and a clear rationale for the study. The introduction section is too long and lacks focus. The authors should provide a more concise and targeted introduction, highlighting the importance and current state of research on the topic. The authors should provide a thorough literature review of previous studies related to GPTMS-TEOS hybrid coatings and the use of DETA as a curing agent and catalyst. Additionally, the objectives of the study should be clearly stated.

3.     The methodology section needs to be more detailed and include information about the specific experimental procedures and conditions used in the study. The authors should describe the synthesis of the GPTMS-TEOS hybrid coatings and the different DETA contents used.

4.     The use of jargon such as "simultaneously promoting the silane network structure and polymer binding affinity" could be made more accessible to a general audience. It is recommended to rephrase the sentence or provide additional explanation.

5.     The authors have provided a detailed analysis of the structural and morphological characteristics of the GPTMS-TEOS hybrid coatings. However, the discussion on the implications of the findings is limited. The authors should expand on the significance of the findings and their potential impact on the performance of the hybrid coatings.

6.     Kindly provide raw data for Figure 2.

7.     A precise description for SEM and TEM images should be provided with the encircle of the major attained findings and same should be described by the authors.

8.     Author needs to reperform the electrochemical test specifically the patterns in Nyquist plot and bodes plot are not acceptable in current medium and surrounding.

9.     Chi square values should be provided in Table 2.

10.  Why there is no discussion about the adsorption and kinetic behavior of the studied inhibitor?

11.  Additionally, the authors should provide more information on the TEM analysis, such as the magnification and scale bar used. This would help readers better understand the size and distribution of the silica spheres in the composite.

12.  The authors should provide more explanation of the TGA and EIS results, including the significance of the data presented in the figures and tables.

13.  The paper should address the language and writing issues. The authors should use clear and concise language, avoid jargon and technical terms that are not defined, and proofread the paper for grammar and spelling errors. Also, some recent literature need to be included such as Journal of Physics: Conference Series (Vol. 2267, No. 1, p. 012079). IOP Publishing, Current Nanoscience, 18(2), 203-216

14.  The conclusions should be revised to provide a clear summary of the main findings and their implications. The authors should also discuss the limitations of the study and suggest directions for future research.

Minor editing of English language required

Author Response

  1.  The abstract is too brief and lacks important details. It would be helpful to provide more context and background information about the study, as well as a brief summary of the main findings and implications.

Response: The abstract was extended somehow to demonstrate the background of the study.

  1. The introduction should be expanded to provide more background information and a clear rationale for the study. The introduction section is too long and lacks focus. The authors should provide a more concise and targeted introduction, highlighting the importance and current state of research on the topic. The authors should provide a thorough literature review of previous studies related to GPTMS-TEOS hybrid coatings and the use of DETA as a curing agent and catalyst. Additionally, the objectives of the study should be clearly stated.

Response: Actually, the present investigation is a little bit new and up to now we could not find some papers that used GPTMS-TEOS and DETA as the starting materials. We only stated the important of corrosion protection by the coatings in the first paragraph. The sol-gel technique was introduced in the second paragraph. The present coating was described in the third of the introduction.

  1. The methodology section needs to be more detailed and include information about the specific experimental procedures and conditions used in the study. The authors should describe the synthesis of the GPTMS-TEOS hybrid coatings and the different DETA contents used.

Response: The dosage of each raw material, the type of preparation vessel, and reaction time and temperature were provided in detail in the manuscript.

During the preparation, the dosage of DETA used in each run was listed in Table 1, and the obtained electrochemical parameters for each run was listed in Table 2.

  1. The use of jargon such as "simultaneously promoting the silane network structure and polymer binding affinity" could be made more accessible to a general audience. It is recommended to rephrase the sentence or provide additional explanation.

Response: This jargon was used in the second paragraph of the Introduction. We already modified the sentence as per the suggestion of the reviewer.

  1. The authors have provided a detailed analysis of the structural and morphological characteristics of the GPTMS-TEOS hybrid coatings. However, the discussion on the implications of the findings is limited. The authors should expand on the significance of the findings and their potential impact on the performance of the hybrid coatings.

Response: The improvement of the coating performance was achieved by employing the organic-inorganic hybrid material prepared using the sol-gel technique. Different dosage of the DETA was examined, as listed in Table, while in Sec. 3.2 the structural and morphological characteristics of the hybrid coatings was investigated. The performance of the hybrid coating was tested in Sec. 3.3 to 3.5.

  1. Kindly provide raw data for Figure 2.

Response: We could provide the raw data of Fig. 2 as per request. Our email number is: [email protected]

  1. A precise description for SEM and TEM images should be provided with the encircle of the major attained findings and same should be described by the authors.

Response: We only provided one SEM images and one TEM images of GPTMS-TEOS composites, as shown in Fig. 3.

  1. Author needs to reperform the electrochemical test specifically the patterns in Nyquist plot and bodes plot are not acceptable in current medium and surrounding.

Response: For the third comment raised by the second reviewer, we already answered the comment. We performed the EIS test, and also carried out a salt spray test. The good protection effect was observed, and hence the polarization test was not presented in the manuscript.

  1. Chi square values should be provided in Table 2.

Response: Thanks to the suggestions of the reviewers, chi-square test is mainly used to study the relationship between categorical data. The data in Table 2 are the equivalent circuit fitting results of bare steel and each coating. The data are relatively small, the correlation between each coating is uncertain, and the significance of each electrochemical parameter is different. Therefore, it is difficult to use chi-square test for table data.

  1. Why there is no discussion about the adsorption and kinetic behavior of the studied inhibitor?

Response: Sorry, we do not understand the meaning of inhibitor. Is it possible for you to describe the comment in detail ?

  1. Additionally, the authors should provide more information on the TEM analysis, such as the magnification and scale bar used. This would help readers better understand the size and distribution of the silica spheres in the composite.

Response: Unlike the method used by other researchers, the hybrid suspension was prepared by the sol-gel method in the present investigation, and hence the size of the nanosized silica was very small. We just paid our attention to the performance of the hybrid coating.

  1. The authors should provide more explanation of the TGA and EIS results, including the significance of the data presented in the figures and tables.

Response: The explanation of the TGA and EIS results was revised in the revised manuscript, as per the request of other reviewers. Some of the figures were redrawn also.

  1. The paper should address the language and writing issues. The authors should use clear and concise language, avoid jargon and technical terms that are not defined, and proofread the paper for grammar and spelling errors. Also, some recent literature need to be included such as Journal of Physics: Conference Series (Vol. 2267, No. 1, p. 012079). IOP Publishing, Current Nanoscience, 18(2), 203-216

响应:手稿经过了大量修改。我们还请一位在英国生活超过25年的朋友提高英语水平。建议的近期文献在修订后的手稿中被引用。

  1. 应修订结论,以明确概述主要调查结果及其影响。作者还应讨论该研究的局限性,并为未来的研究方向提出建议。

响应:修改了结论的部分。描述了涂层的局限性,并指出了未来的工作。

Round 2

Reviewer 1 Report

Find below comments which must be considered to improve the quality of the presented paper:

Lines 11, 43, 63, 100 – Correct the nomenclature of the chemical compounds according to literature: 3-glycidoxypropyltrimethoxysilane (GPTMS) and tetraethylorthosilicate (TEOS) (without spaces).

Lines 74 – Table 1 should not be placed in the Introduction section, it is more related to Materials preparation section.

Lines 95 – Correct the nomenclature of SiO2.

Lines 95-97 – Attention: The law of thermal stability of hybrid coatings was studied and discovered”. “Discover the best mixing ratio and the best corrosion resistance.”

      A law involves to present the law equations and fit these equations to the results explaining it. Also, the last phrase “Discover the... has no meaning.  The sentence of lines 95-97 need be rewritted.

Attention: Correct the lack of spaces in legends of axes in Figures 1-5.

Line 117 – Thickness of the coatings are reported to have 80 nm, but authors do not report how it was measured, neither present any measurement. It is known that GPTMS-TEOS hybrids are viscous solutions that yield films with thicknesses of micrometers. Examining the bottom of Fig. 7, the solidified material accumulated at the end of the metallic substrate gives idea how thick the films are. Only report the thickness if have the measurement, and is needed to describe the equipment used.

Line 210 – Legend of Fig. 5 do not mention the inset in graph B. Description of inset should appear in the figure legend.

Line 282 – Attention: In the scheme presented (Fig. 6), authors must inform “who are” the moieties that connect the large spheres of SiO2. Glycidoxypropyl will polymerize forming cross-linked species. See the work of Plinio Innocenzi et al., Sol-gel reactions of 3-glycidoxypropyltrimethoxysilane in a highly basic aqueous solution, Dalton Trans., (2009), 9146–9152 DOI: 10.1039/b905830c.

Line 209-210: Figure 5 is an very important result. Legends A-F can be elegantly placed inside the graphs, so these graphs A-D can have increased size and be better arranged in the page.

Lines 11, 43, 63, 100 – Correct the nomenclature of the chemical compounds according to literature: 3-glycidoxypropyltrimethoxysilane (GPTMS) and tetraethylorthosilicate (TEOS) (without spaces).

Lines 95 – Correct the nomenclature of SiO2.

Attention: Correct the lack of spaces in some text sentences.

Author Response

1、Lines 11, 43, 63, 100 – Correct the nomenclature of the chemical compounds according to literature: 3-glycidoxypropyltrimethoxysilane (GPTMS) and tetraethylorthosilicate (TEOS) (without spaces).

Response: Thank you for giving the kind suggestions. Under your guidance, the error was corrected.

2、Lines 74 – Table 1 should not be placed in the Introduction section, it is more related to Materials preparation section.

Response: Thank you for giving the kind suggestions. Under your guidance, Table 1 is moved to the coating preparation section of the second part。

3、Lines 95 – Correct the nomenclature of SiO2.

Response: Thank you for giving the kind suggestions. Under your guidance, the error was corrected.

4、Lines 95-97 – Attention: “The law of thermal stability of hybrid coatings was studied and discovered”“Discover the best mixing ratio and the best corrosion resistance.”

      A law involves to present the law equations and fit these equations to the results explaining it. Also, the last phrase “Discover the...“ has no meaning.  The sentence of lines 95-97 need be rewritted.

Response: Thank you for giving the kind suggestions. Under your guidance, The sentence of lines 95-97 has been revised.

The thermal stability of hybrid coatings was studied and the best dosage of the curing agent (DETA) was obtained.

5Attention: Correct the lack of spaces in legends of axes in Figures 1-5.

Response: Thank you for giving the kind suggestions. Under your guidance, the error was corrected.

6、Line 117 – Thickness of the coatings are reported to have 80 nm, but authors do not report how it was measured, neither present any measurement. It is known that GPTMS-TEOS hybrids are viscous solutions that yield films with thicknesses of micrometers. Examining the bottom of Fig. 7, the solidified material accumulated at the end of the metallic substrate gives idea how thick the films are. Only report the thickness if have the measurement, and is needed to describe the equipment used.

Response: Thank you for giving the kind suggestions, The use of ultrasonic thickness gauges to measure coatings was added to the paper.

7、Line 210 – Legend of Fig. 5 do not mention the inset in graph B. Description of inset should appear in the figure legend.

Response: The inset in the graph B of Fig. 5 was described in line 210 of Sec. 3.3 in the revised manuscript.

8、Line 282 – Attention: In the scheme presented (Fig. 6), authors must inform “who are” the moieties that connect the large spheres of SiO2. Glycidoxypropyl will polymerize forming cross-linked species. See the work of Plinio Innocenzi et al.Sol-gel reactions of 3-glycidoxypropyltrimethoxysilane in a highly basic aqueous solution, Dalton Trans., (2009), 9146–9152 DOI: 10.1039/b905830c.

Response: In lines 282-286 of the manuscript, a reference to this has been added.

9、Line 209-210: Figure 5 is an very important result. Legends A-F can be elegantly placed inside the graphs, so these graphs A-D can have increased size and be better arranged in the page.

Response: Thank you for giving the kind suggestions. Due to the limitation of the journal publishing format, the layout of the figure should be typeset in accordance with the typesetting requirements of the caotings journal, so the A-D in Figure 5 can only be typeset according to the current format, and I will try to enlarge the A-D figure.

Reviewer 4 Report

The manuscript reports on the structural, and electrochemical properties of anti-corrosion silica-epoxy coatings, based on TEOS, GPTMS, and DETA precursors, to protect Q215 steel. The authors addressed a part of the referee´s comment, the following points might be addressed to improve the work.

Regarding:

- Comment 8: Inform the method used for film thickness measurements.

- Comment 12 was not satisfactorily responded (Authors neither include correction performed in the manuscript nor inform the page of where the changes were performed).

- Comment 14: Provide adhesion strength to the steel substrate, using a standard ANSI method.

Moderate editing of English language required

Author Response

1、- Comment 8: Inform the method used for film thickness measurements.

Response: Thank you for giving the kind suggestions. Under your guidance, The use of ultrasonic thickness gauges to measure coatings was added to the manuscript。

2、- Comment 12 was not satisfactorily responded (Authors neither include correction performed in the manuscript nor inform the page of where the changes were performed).

Response:Thank you very much for the reminder, we have already responded in response to the review, missing changes in the manuscript, and this time we have revised the questions.

3、- Comment 14: Provide adhesion strength to the steel substrate, using a standard ANSI method.

Response:The adhesion strength test of the coating in this paper adopts the lattice method, and the adhesion strength of the coating prepared by the experiment is better, but due to the length of the manuscript, the adhesion strength test part is not reflected in the manuscript. The later research team will publish further papers on related aspects